# Functional and Combined Training Promote Body Recomposition and Lower Limb Strength in Postmenopausal Women: A Randomized Clinical Trial and a Time Course Analysis

**DOI:** 10.3390/healthcare12090932

**Published:** 2024-05-01

**Authors:** Marcos Raphael Pereira-Monteiro, José Carlos Aragão-Santos, Alan Bruno Silva Vasconcelos, Antônio Gomes de Resende-Neto, André Filipe Santos de Almeida, Luis Alberto Gobbo, Francisco Hermosilla-Perona, Juan Ramón Heredia-Elvar, Fabricio Boscolo Del Vecchio, Felipe J. Aidar, Marzo Edir Da Silva-Grigoletto

**Affiliations:** 1Graduate Program in Physiological Sciences, Federal University of Sergipe, São Cristóvão 49100-000, Sergipe, Brazil; abs.vasconcelos@gmail.com (A.B.S.V.); fjaidar@academico.ufs.br (F.J.A.); medg@ufs.br (M.E.D.S.-G.); 2Graduate Program in Health Sciences, Federal University of Sergipe, Aracaju 49060-676, Sergipe, Brazilafilipe@academico.ufs.br (A.F.S.d.A.); 3Department of Physical Education, Estácio de Sergipe University Center, Aracaju 49020-530, Sergipe, Brazil; 4Department of Physical Education, São Paulo State University, Presidente Prudente 19060-900, São Paulo, Brazil; luis.gobbo@unesp.br; 5Facultad de Ciencias de la Vida y la Naturaleza, Universidad Nebrija, 28015 Madrid, Spain; fperoher@uax.es; 6Department of Physical Activity and Sports Science, Alfonso X El Sabio University, 28691 Madrid, Spain; jelvaher@uax.es; 7School of Physical Education, Federal University of Pelotas, Pelotas 96055-630, Rio Grande do Sul, Brazil; fabricioboscolo@ufpel.edu.br; 8Graduate Program in Physical Education, Federal University of Sergipe, São Cristóvão 49100-000, Sergipe, Brazil

**Keywords:** aging, body composition, exercise, health

## Abstract

Encouraging healthy aging in postmenopausal women involves advocating for lifestyle modifications, including regular physical exercise like combined training (CT) and functional training (FT). Regarding this population, age-related alterations in body composition, such as decreased muscle mass and heightened adipose tissue, impact health. The aim of this study was to analyze the effects of FT and CT on body recomposition in postmenopausal women. About the methods, we randomly allocated 96 post-menopausal women to the FT, CT, or control group (CG). We measured body composition by bioimpedance and lower limb muscle strength by sit-to-stand test in five repetitions, respectively. The training protocol lasted 16 weeks, and we measured body composition and lower limb muscle strength every 4 weeks, totaling five assessments. Regarding results, we notice that both training groups increased lean mass from the 8th week of training. In addition, a reduction was observed in total fat percentage and an increase in appendicular lean mass from the 12th week of intervention. No differences were found for body mass. Furthermore, only the experimental groups increase muscle strength, starting from the 4th week of training. The conclusion was that FT and CT promote similar adaptations in body recomposition without affecting body mass in postmenopausal women.

## 1. Introduction

Among the physical capacities that are affected by aging, declines in strength and cardiorespiratory capacity have a direct negative impact on health and functionality, and are associated with pathological conditions such as sarcopenia and cardiovascular disease [1,2]. In addition, balance, coordination, flexibility, and muscle power are also reduced during the aging process; these are essential for a low risk of falls and maintaining activities of daily living [3,4]. To promote healthy aging, the current literature encourages changes in lifestyle habits, with emphasis on regular physical exercise [5,6,7]. Therefore, exercise has been used to reduce the effects of senescence and increase longevity in postmenopausal women, either as resistance or aerobic training methods [8,9].

Currently, training proposals that stimulate more than one physical capacity have been widely used, such as combined training (CT) and functional training (FT) [10,11]. Specifically, CT uses two different training approaches in the same session; commonly, these are strength training and endurance training, prioritizing stimulus in strength and aerobic capacity [12]. Meanwhile, FT is characterized as multicomponent training, addressing different capacities such as strength, coordination, flexibility, power, aerobic endurance, and balance in the same session [13]. These types of training stand out for their ability to promote functionality, autonomy, and body composition, mainly in postmenopausal women [14,15].

In this specific populational group, the aging-associated changes in body composition, such as the reduction in muscle mass and increases in adipose tissue [16,17], are more pronounced due to the postmenopausal estrogen drop [18,19]. In turn, body composition stands out as a good health indicator related to postmenopausal women’s health. Additionally, body composition changes are related to inflammatory status [20,21], longevity [22], physical disability [23], and mortality [24,25]. Corroborating this, Bosch et al. (2015) showed that the fat percentage of 38.3% for women and 23.4% for men represent thresholds in adiposity. Above this percentage, increases are associated with increased cardiovascular risk and insulin resistance [26]. Increases in fat percentage in older people are related to fat infiltration in muscle tissue, negatively impacting the neuromuscular function and, consequently, the functionality and health of older people [27,28].

Among the training proposals, FT and CT presented good impacts in the body composition, increasing lean mass and reducing body fat [29,30]. However, there are discrepancies in the literature results, possibly due to different intervention proposals, training doses, and measurement methods [29,31,32,33,34]. Specifically with regard to FT, the fact that it is presented as multicomponent training with exercises that mimic functionality brings it closer to the proposals presented in the latest guidelines for training older people [11,15]. However, these characteristics reinforce the need for more technical details about the methods used. Furthermore, there are no studies that show these improvements on body composition in a simultaneous way, reducing body fat and increasing muscle mass in postmenopausal women. One of the main limitations of the studies is probably the duration of exposure to physical exercise, with shorter interventions lasting up to 12 weeks [29,30].

To improve health, the events of simultaneously increasing lean mass and reducing body fat receive the name of body recomposition [35]. To verify this phenomenon, it is necessary to understand how long it takes for FT and CT to promote changes in lean mass and body fat in postmenopausal women, and if these changes appear in a simultaneous way. The literature points out that the sequence of adaptations to resistance training involves improvements in lean mass increase, directly associated with hypertrophy, starting around 8 weeks of training [36]. Regardless of adaptations in muscle mass, there have already been improvements in muscle strength in the initial weeks of the training, this being related to neural adaptations of exercise [37]. In this way, it is important to verify the impact of these morphological adaptations on the muscle strength.

This gap stems from the fact that most studies only perform assessments at the beginning and end of interventions; this design may impede the identification of valuable information for this population, such as the minimum time required for significant adaptations to occur. In this view, a time course design can provide insight into the visualization of these variables. Additionally, FT and CT may be an appropriate option to mitigate the effects of menopause on body composition, strength, and functionality. In view of this scenario, understanding the timing and simultaneity of adaptations in body composition promoted by CT and FT in older women can guide the practice of movement professionals and the application of these protocols in public health policies. Therefore, the objective of this study was to analyze and compare the effects of FT and CT on the body recomposition of postmenopausal women. Our hypothesis is that 16 weeks of both training methods could increase lean mass and decrease fat mass in postmenopausal women in a simultaneous way, in addition to promoting strength and functionality in a similar way.

## 2. Materials and Methods

### 2.1. Experimental Design

This study is a randomized controlled clinical trial, registered in the Clinical Trials Registry at clinicaltrials.gov (RBR-2d56bt) and approved by the Ethics Committee of the Federal University of Sergipe (protocol code 3.225.938; approval date: 27 March 2019). It was applied with one protocol of FT and another of CT (independent variables) and evaluated the body composition and lower limb muscle strength every 4 weeks (dependent variables) in postmenopausal women. The intervention featured 2 weeks of participant familiarization with the intervention protocols and another 16 weeks committed to the training protocols. The participants were submitted to five moments of evaluation. Evaluations of body composition and lower limb muscle strength were performed in all moments (W0, W4, W8, W12, and W16) and conducted in the physiology laboratory of the Federal University of Sergipe (Figure 1). Recruitment was conducted from March 2019 and the follow-up period was between April and October 2019. This study report followed the CONSORT 2010 checklist (Appendix A).

### 2.2. Participants

The recruitment was carried out through dissemination on social media and leafleting around the Federal University of Sergipe, as well as by attending public markets and religious events. We calculated the sample size in G*Power software (version 3.1.9.2, Kiel, Germany) based on the observed changes in fat percentage by Resende-Neto et al. (2019a) [31]. Specifically, adopting an alpha of 0.05 and power of 80%, 27 participants per group were required. We included 32 participants per group, increasing the sample size by 20% assuming their eventual loss throughout the intervention. A total of 200 volunteers reported to the laboratory. Considering the inclusion criteria necessary for participation in the study (60 years old or older; female; have no menstrual bleeding in the past 12 months; have no musculoskeletal or cardiovascular contraindications to the practice of high-intensity physical training; have not exercised regularly for at least 6 months), 96 participants were randomly distributed to one of the following groups: functional training (FT); combined training (CT); and control group (CG) (Figure 2). In the case of participation in another systematized exercise program, non-attendance in three consecutive sessions, or absence in more than 15% of the training sessions, the participant was removed from the program.

The distribution between groups was conducted using block randomization. Participants were ranked based on body fat percentage, grouped into a block of three, and allocated to each group equally. The distribution process was blind and randomized by an independent researcher.

Additionally, the volunteers were asked to sign an informed consent form and had the risks and benefits of the research explained to them.

### 2.3. Intervention Protocols

The protocols consisted of 44 training sessions, conducted from 6 to 8 a.m., lasting 45 min, performed on non-consecutive days over 16 weeks, three times per week, by the FT and CT groups. During the familiarization weeks, all the subjects participated in two sessions of each protocol (FT, CT and CG). In these sessions, the main intention was for all of them to learn the exercises used. In this sense, aspects of intensity were adjusted to enable learning.

When the training protocol began, the participants performed all the exercises at a maximum concentric speed. To control the external load, we maintained a range between 8 and 12 repetition maximum (RM) during the exercises, with adjustments in the external load (of 5 or 10%) or complexity of the exercises performed, when necessary (i.e., when the participant performed 7 repetitions or fewer in the proposed time, the external load or complexity was reduced. On the other hand, when the participant performed 13 repetitions or more, the external load or complexity was increased). We monitored the participants through the rate of perceived exertion scale (RPE), which was answered after each session of training [38]. We used the RPE because of its ease of application in large groups. For both experimental groups (FT and CT), progressions based on complexity were used from the 8th week of training onwards.

The intervention protocols were performed at the gym of the Federal University of Sergipe. Trained professionals supervised the participants (1:6) to ensure the performance and safety of the exercises. Additionally, we oriented the participants to maintain food habits and daily activity habitually.

#### 2.3.1. Functional Training

As shown in Table 1, the exercise protocol was composed of four parts aimed at different physical capacities, as follows: Part 1, 3 min of preparing for movement; Part 2, 10 min of exercises focused on muscle power, agility, and speed; Part 3, 10 min of exercise aimed at muscle strength in functional movement patterns, such as squatting, pushing, pulling, and carrying; and Part 4, 10 min of intermittent aerobic exercises [32].

#### 2.3.2. Combined Training

Similar to the FT protocol, there were four distinct parts used in the CT protocol, as shown in Table 1: Part 1, 3 minutes of general and specific warm-up; Part 2, 16 min of exercises focused on muscle strength realized on typical gym machines; Part 3, 10 min of exercises aimed at cardiorespiratory fitness; and Part 4, stretching exercises for the major muscle groups [39].

#### 2.3.3. Control Group

The control group performed submaximal static stretching exercises with two sets of 15 s for the major muscle groups and meditation practices. These exercises were performed in 45 min sessions, three times per week, for 16 weeks.

### 2.4. Data Collection Procedure

Data collection included outcome variables (body composition and lower limb muscle strength) and sample characterization variables; these were obtained from anthropometric scales (Líder^®^, P150C, São Paulo, Brazil) and the anamnesis of each participant. The procedures were conducted by previously trained evaluators, who performed the same tests during the whole study, and were blinded to all experimental groups.

#### 2.4.1. Body Composition

We evaluated the body composition with an octapolar electric bioimpedance scale (Tanita BC-558 Ironman^®^, Tokyo, Japan). This bioimpedance system uses an electric current with a frequency of 50 kHz that measures the amount of intracellular and extracellular water. This system estimates values referring to total body fat and lean mass, appendicular lean mass, and body mass, subsequently for analysis [40,41].

The measurement was standardized according to the manufacturer’s recommendations: fasting for at least 6 h; no physical exercise on the day before the test; no caffeine, alcohol or diuretic drinks within 24 h before the evaluation; and normal water intake [42]. The participants were instructed to follow the recommendations prior to assessment. We only performed the evaluation if we could confirm that the participant followed all the recommendations. If the required protocol was not confirmed, the participant would undergo the assessment on the following day.

The intraclass correlation coefficient (ICC) for the use of this equipment was previously calculated with 31.25% of the sample (*n* = 30) and presented a value of 0.91, considered excellent [43]. On the other hand, regarding validity, this equipment presents an intermethods agreement considered very high with the Dual-energy X-ray absorptiometry, being for fat mass, r = 0.95, and for lean mass, r = 0.93 [41].

#### 2.4.2. Lower Limb Muscle Strength

We evaluated the lower limb muscle strength of the sample in a test that reproduces its functionality. For the five times sit-to-stand test (FTSTS), the individual must perform five repetitions of sitting down and getting up from a chair (45 cm, fixed base, AT51, Araquari, Santa Catarina, Brazil) as quickly as possible. The individual’s positioning should involve the arms crossed over the trunk. The test is counted in time, starting with the evaluator’s verbal command (now!) and ending with the performance of the last repetition. This test has excellent reproducibility with an ICC of 0.95 [44].

### 2.5. Statistical Analysis

For the categorical variables, the values were expressed as relative and absolute frequency, and the Chi-Square test was used to verify the differences between the groups. For the numerical variables, the values were expressed as mean and standard deviation, had their normality checked by the Shapiro–Wilk test, and one-way Analysis of Variance (ANOVA) was used to verify the differences between the groups.

For the outcome variable values, these data were expressed as mean and standard deviation and analyzed using the Shapiro–Wilk test for normality and Levene’s test for homogeneity. We used an intention-to-treat analysis. Then, we used a repeated-measures ANOVA with two factors (time x group) and Bonferroni post hoc. A Mauchly’s test of sphericity was used to test this assumption, and a Greenhouse–Geisser correction was used when necessary. For all group and time comparisons, we calculated the effect size (ES) using the methodological procedures defined by Cohen [45] and percentage change (Δ). All tests were two-tailed, and we adopted *p* < 0.05. Experienced researchers performed the statistical processing using Jamovi (The jamovi project, Sydney, Australia, version 2.3.28.

## 3. Results

The groups showed no statistical differences at W0 (Table 2). Furthermore, throughout the 16 weeks of training, there was a sample loss of 3.13% in the FT group, 15.63% in the CT group, and 12.5% in the CG group (Figure 2).

We detected a time effect for all variables (Figure 3), with the exception of the body mass, that none of the groups showed changes over time (F (GL) = 0.212; *p* = 0.334; ƞ^2^ = 0.000) (Figure 3A). Regarding the fat percentage, experimental groups showed significant decrease at W12 compared to W0 (FT: Δ = −3.9%, ES = −0.30, *p* = 0.002; CT: Δ = −4.8%, ES = 0.31, *p* < 0.001), maintaining such differences also at W16 compared to W0 (FT: Δ = −5.2%, ES = 0.40, *p* < 0.001; CT: Δ = −5.8%, ES = 0.38, *p* < 0.001) (Figure 3B).

Regarding lean mass, compared to W0 significant increases were noted for the FT and CT groups at W8 (FT: Δ = +2.0%, ES = 0.16, *p* = 0.028; CT: Δ = +2.1%, ES = 0.19, *p* = 0.048), W12 (FT: Δ = +3.2%, ES = 0.27, *p* = 0.002; CT: Δ = +3.6%, ES = 0.34, *p* < 0.001), and W16 (FT: Δ = +3.7%, ES = 0.31, *p* < 0.001; CT: Δ = +4.3%, ES = 0.41 *p* < 0.001) (Figure 3C). As for appendicular lean mass, the data show increases for the FT and CT groups starting at W12 (FT: Δ = +3.4%, ES = 0.25, *p* = < 0.001; CT: Δ = +2.3%, ES = 0.19, *p* = 0.031), and being maintained at W16 (FT: Δ = +3.4%, ES = 0.25, *p* < 0.001; CT: Δ = +2.3%, ES = 0.19, *p* = 0.019) (Figure 3D).

For lower limb muscle strength and functionality, using the FTSTS data we found a statistically significant reduction in the time needed to perform the task starting at W4 for the FT and CT, maintaining such differences until the end of the intervention. The CG showed no statistically significant differences over time (Table 3).

## 4. Discussion

Our main findings confirm our initial hypothesis since the FT and CT promoted similar increases in lean mass and similar reductions in the percentage of fat in postmenopausal women in a simultaneous way. To the best of our knowledge, this is the first study that targeted body recomposition with these training proposals in this population. For both training groups, the increase in lean mass was detected in the 8th week of training, while the reduction in fat percentage was in the 12th week. These changes in body composition were maintained at the end of the intervention period. Thus, our findings point to the efficacy of both training methods applied to the promotion of body recomposition in postmenopausal women and provide subsidies for the performance of exercise professionals.

The body composition changes observed in the present study reinforce previous findings regarding the benefits of exercise on this variable [46,47]. Resende-Neto et al. (2019) approached a model of FT and CT similar to the one adopted in the present study, finding fat percentage decreases after the application of FT and lean mass increases after a CT protocol in older women, both for 12 weeks [31]. These data should be interpreted with caution since the study does not present a proper equalization of training volume between groups. In addition, it is only 12 weeks long, and only the pre- and post-intervention periods were evaluated, not allowing for the verification of the simultaneity of changes in the variables. Possibly, the similar adaptations between FT and CT in our study were derived from the volume equalization, since the improvements in body composition appear to be related to this variable [48,49].

We observed an increase in lean mass from 8 weeks for both interventions; specifically, on the adaptations in appendicular lean mass, our study observed increases from 12 weeks of intervention. A possible mechanism for this is the sequence of adaptations provided by resistance training, starting with neural adaptations that occurred in the initial weeks of training, followed by muscular adaptations involving hypertrophy and changes in muscle architecture, which are commonly reported after 8 weeks of training [50,51]. Our data regarding muscle strength reinforce this theory since improvements in lower limb muscle strength were found after the first 4 weeks, when there were still no significant differences in lean mass, possibly due to neural adaptations promoted by the exercise. Similarly, Nascimento et al. (2019) performed 12 weeks of resistance training in older women and found increases in lean mass and appendicular lean mass [52]. Thus, the importance of increasing muscle mass to decrease the risk of mortality and counteracting the process of sarcopenia in older people should be emphasized [53].

Our sample presented body fat reduction after 12 weeks of intervention. This fat reduction can be explained by the energy expenditure provided by physical exercise [54]; it is important to understand the time aspects that involve this reduction. Timmons et al. (2018) showed reduced body fat in older people after 6 weeks of aerobic training or CT; however, the resistance training group only showed this adaptation after 12 weeks of intervention [30], demonstrating that different types of training require different periods to provide adaptations in body fat. It is worth pointing out that high levels of body fat are related to the onset of cardiovascular diseases [26,55].

Despite improvements in body composition, the absence of differences in body mass can be explained by the possible offset between increased lean mass and reduced adipose tissue at the same time, causing the phenomenon of body recomposition [35]. We can notice that both training protocols promoted body recomposition in our study. Without presenting effects on body mass, the proposed interventions acted simultaneously on the percentage of fat and lean mass throughout the time course. The same phenomenon was seen in a previous study that used resistance training in healthy men and women [56]. These data reinforce the importance of specifically analyzing total body fat and lean mass measurements for a more accurate interpretation of body composition in this population, suggesting the replacement of more generalist markers such as body mass index (BMI) [57].

Some limitations of our study are that there was no nutritional and physical activity control of the sample; however, we instructed the participants to maintain the same eating and physical activity habits throughout the intervention period. Furthermore, Swift et al. (2018) demonstrated that exercise interventions without eating control can reduce body fat and increase lean mass [58]. In this way, important studies have been published about body composition adaptations through physical training without nutritional and physical activity control [59,60], which tend to approximate more closely to conditions resembling real-world scenarios.

Thus, our study brings out several positive points concerning the current literature, such as the time-course of the adaptations promoted by exercise on body composition. In addition, it is noteworthy that the functional training modality has proven to be more attractive in our study, because in this group there was greater adherence to this proposal of training. One possible explanation lies in the fact that functional training has a more dynamic approach, providing greater enjoyment levels in training sessions than traditional training [61]. From a practical point of view, our study provides data on two different methods of training, both of easy applicability in public health policies, that promote body recomposition in postmenopausal women. In this way, our study provides important information for professional practice and the setting up of public health policies.

## 5. Conclusions

Sixteen weeks of FT and CT have similar effects in promoting body recomposition, increasing muscle mass from the 8th week of training and reducing body fat from the 12th week of training. In addition, both proposals of training promote increases in lower limb muscle strength in postmenopausal women from the 4th week of training. Thus, FT and CT present as stimulating alternatives beyond traditional resistance or aerobic training, for health promotion in this public, promoting functionality through a multicomponent proposal.

## Figures and Tables

**Figure 1 healthcare-12-00932-f001:**
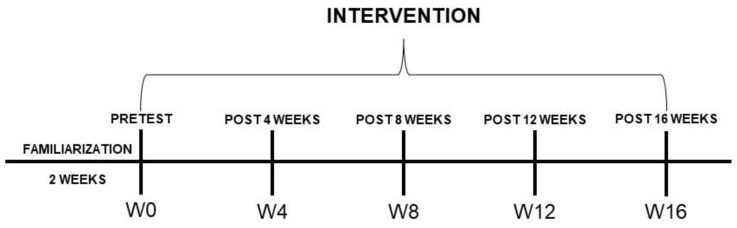
Experimental design. Notes: W0: pre-intervention moment; W4: moment after 4 weeks of intervention; W8: moment after 8 weeks of intervention; W12: moment after 12 weeks of intervention; W16: moment after 16 weeks of intervention.

**Figure 2 healthcare-12-00932-f002:**
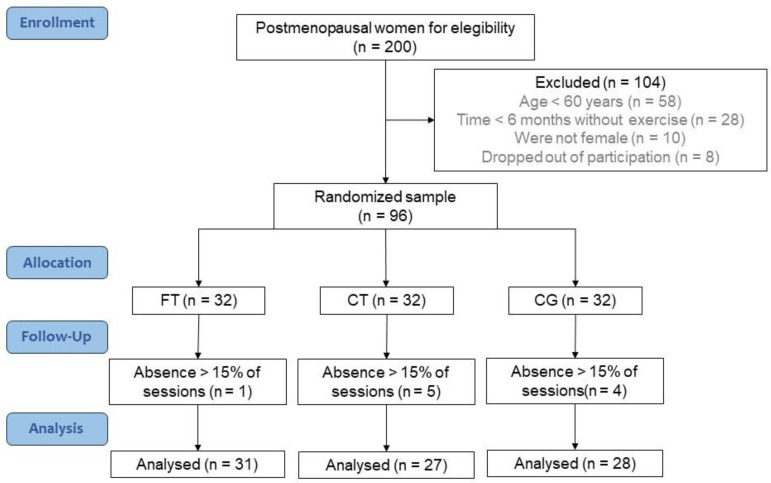
Flowchart. Abbreviations: FT: Functional Training. CT: Combined Training. CG: Control Group.

**Figure 3 healthcare-12-00932-f003:**
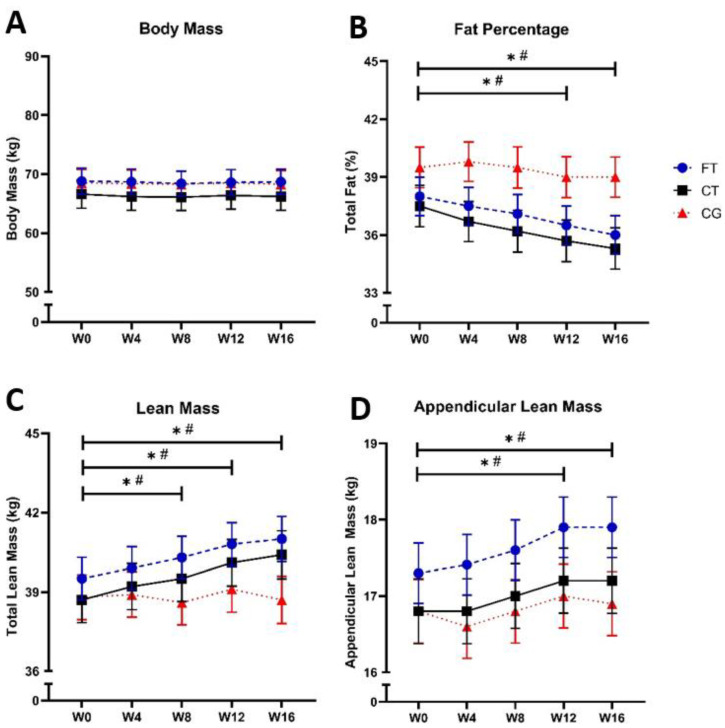
Observed results on body composition variables. Body Mass (**A**). Fat Percentage (**B**). Lean Mass (**C**) and Appendicular Lean Mass (**D**) over 16 weeks of intervention. Note. *: effect of time relative to FT. #: effect of time relative to CT. Values are expressed as mean and standard error mean. Statistical difference verified by repeated-measures ANOVA. Abbreviations: FT: Functional Training. CT: Combined Training. CG: Control Group.

**Table 1 healthcare-12-00932-t001:** FT and CT training protocols.

	PART 1	PART 2	PART 3	PART 4
FUNCTIONAL TRAINING (FT)	Mobility, muscle activation, and motor coordination.	Exercises for muscle power, agility, and balance.	Exercises for strength in functional patterns using free weights.	Interval running.
	Time: 3 min.	Time: 10 min Stations: 5 Passages: 2 Density: 30/30 RPE: 6–7.	Time: 10 min Stations: 8 Passages: 2 Density: 40/40 RPE: 7–9 Intensity: 8–12 RM.	Time: 10 min Density: 40/40 RPE: 6–7.
COMBINED TRAINING (CT)	General and specific warm-up.	Exercises for strength using analytical machines.	Intermittent walking and running.	Active stretching.
	Time: 3 min.	Time: 16 min. Stations: 8. Passages: 2. Density: 40/40. RPE: 7–9. Intensity: 8–12 RM.	Time: 10 min. Density: 40/40. RPE: 6–7.	Time: 5 min. Density: 40/40. RPE: 3–4.

Abbreviations: RM: Repetition Maximum. RPE: Rated Perceived Exertion Scale.

**Table 2 healthcare-12-00932-t002:** Participants’ characterization.

Characteristics	FT	CT	CG	*p*
Anthropometry (mean and standard deviation)				
Age (years)	63.6 ± 3.4	65.2 ± 4.5	67.1 ± 5.8	0.063
Height (m)	1.54 ± 0.06	1.55 ± 0.06	1.53 ± 0.06	0.802
BMI (kg/m^2^)	28.84 ± 4.81	27.80 ± 4.54	29.23 ± 5.28	0.527
Smoking (relative and absolute frequency)				
Smoker	3.2 (1)	0.0 (0)	10.7 (3)	0.362
Ex-smoker	25.8 (8)	18.5 (5)	21.4 (6)	
Never smoked	71.0 (22)	81.5 (22)	67.9 (19)	
Medical History (relative and absolute frequency)				
Hypertension	45.2 (14)	51.9 (14)	57.1 (16)	0.653
Diabetes	19.4 (6)	14.8 (4)	25.0 (7)	0.650
Dyslipidemia	41.9 (13)	51.9 (14)	39.3 (11)	0.613
Alcohol intake	29.0 (9)	25.9 (7)	25.0 (7)	0.934

Abbreviations: BMI: Body Mass Index. FT: Functional Training. CT: Combined Training. CG: Control Group.

**Table 3 healthcare-12-00932-t003:** Observed results on FTSTS for lower limb muscle strength and functionality.

	W0	W4	W8	W12	W16
FT (s)	6.88 ± 1.70	6.01 ± 1.63 *	5.64 ± 1.49 *#	5.28 ± 1.22 *#	5.04 ± 1.06 *#
CT (s)	8.01 ± 1.79	7.19 ± 1.81 *	6.33 ± 1.58 *	5.97 ± 1.37 *#	5.82 ± 1.28 *
CG (s)	7.43 ± 1.66	7.51 ± 1.89	7.96 ± 2.02	7.76 ± 2.02	7.68 ± 2.05

Note. *: effect relative to W0. #: effect relative to CG.

## Data Availability

The data presented in this study are available on request from the corresponding author.

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
