# Peer review of "Functional and Combined Training Promote Body Recomposition and Lower Limb Strength in Postmenopausal Women: A Randomized Clinical Trial and a Time Course Analysis"

_healthcare, 2024, doi:10.3390/healthcare12090932_

Round 1
Reviewer 1 Report
Comments and Suggestions for Authors
Please specify distinctly and clearly in the paper summary: methods, results and discussion, conclusion.
The main objectives or hypotheses of this research do not emerge from the introduction. Please enter in the content of the introduction: objectives or hypotheses.
In the "Discussions" you state that: "Our main findings confirm our initial hypothesis ...) but in the content of the paper you did not present any hypothesis.
You mention that the assessment was conducted at the physiology laboratory of the institution. However, the authors come from five different universities across Brazil and Spain. Which institution?
Author Response
Response: Dear reviewer, thank you for your notes. The alterations provided by our conversation were highlighted in green in the manuscript file.
#Reviewer_1: Please specify distinctly and clearly in the paper summary: methods, results and discussion, conclusion.
Response: Dear reviewer, thank you for your comment. Considering that the journal's rules call for an unstructured abstract (without headings), we have inserted short sentences before each stage in order to resolve this situation. Unfortunately, due to word limitations, we have not inserted the discussion. (lines 23, 24, 28 and 32)
#Reviewer_1: The main objectives or hypotheses of this research do not emerge from the introduction. Please enter in the content of the introduction: objectives or hypotheses.
Response: Dear reviewer, thank you. We have made our objectives and hypotheses clearer in the text (lines 94 – 101)
#Reviewer_1: In the "Discussions" you state that: "Our main findings confirm our initial hypothesis ...) but in the content of the paper you did not present any hypothesis.
Response: Dear reviewer, we believe that with the changes made as a result of the previous question, we have already resolved this issue (lines 94 – 101).
#Reviewer_1: You mention that the assessment was conducted at the physiology laboratory of the institution. However, the authors come from five different universities across Brazil and Spain. Which institution?
Response: Dear, this physiology laboratory is located at the Federal University of Sergipe. We have inserted this information in the text. Thank you. (lines 113 – 114)
We would like to thank you for your contribution and we are grateful for the new version of the article that has been achieved as a result of the changes provided by your comments. We hope that the answers sent, as well as the changes derived from the review process, address your concerns and meet the necessary requirements for publication.
Reviewer 2 Report
Comments and Suggestions for Authors
The present study analyses the effects of combined and functional training in body composition and lower limb strength in Postmenopausal women. Despite the interesting results, there are some concerns that I consider essential to be addressed before the manuscript’s publication.
The title should be more specific. There are analyses that were performed that aren’t contemplated in the title. The lower limb strength, per example.
Introduction
I consider that the introduction section is well-designed, however, we need more info about the physical capacities that will be included in CT and FT. If CT includes aerobic and resistance, we need info about these physical capacities and why it is positive for this population.
In Line 54, the authors identified, “Increases in fat percentage in older people is related to fat infiltration in muscle tissue, impacting the neuromuscular function and, consequently, the functionality and health of older people [21,22].” – It is important identify if it impacts positively or negatively.
The authors also studied the effect of CT and Ft in lower limb strength, however, there isn’t any justification for it in the introductory section.
Methods
The authors identified in the intervention calendar a familiarisation phase. It will be interesting to identify what the authors did in this phase.
Line 115 - These are the conditions for women to be in the menopausal phase. These criteria are indicated for whom?
Line 137 – “The participants performed all exercises at maximum concentric speed. To control the external load, we maintained a range between 8 and 12 repetition maximum (RM)…” 8 and 12 repetitions maximum? Please, clarify.
Line 144 – “The intervention protocols were performed at the gym of the institution.” What institution?
Line 162 – Why the control group performed submaximal static stretching exercises with two sets of 15 seconds for the major muscle groups and meditation practices?
Results
Line 227 - It is the first time that the term appendicular lean mass appears. Please, uniform all document.
Discussion
It will be very interesting to have some discussion about the identical variations between FT and CT.
Conclusion
Line 317 – Please, fix - similar cts in.
Author Response
#Reviewer_2: The present study analyses the effects of combined and functional training in body composition and lower limb strength in Postmenopausal women. Despite the interesting results, there are some concerns that I consider essential to be addressed before the manuscript’s publication.
Response: Dear reviewer, thank you for your notes. The alterations provided by our conversation were highlighted in green in the main document
#Reviewer_2: The title should be more specific. There are analyses that were performed that aren’t contemplated in the title. The lower limb strength, per example.
Response: Dear Reviewer, thank you for your comment. We have changed the title to be more specific (line 3).
#Reviewer_2: Introduction
I consider that the introduction section is well-designed, however, we need more info about the physical capacities that will be included in CT and FT. If CT includes aerobic and resistance, we need info about these physical capacities and why it is positive for this population.
Response: Dear Reviewer, thank you for your insightful comment. We have made some adjustments in the text that hopefully will address your concerns. (lines 38 – 41 and 47 – 55)
#Reviewer_2: In Line 54, the authors identified, “Increases in fat percentage in older people is related to fat infiltration in muscle tissue, impacting the neuromuscular function and, consequently, the functionality and health of older people [21,22].” – It is important identify if it impacts positively or negatively.
Response: Thank you for your carefully reading. We’ve changed the sentence (line 65).
#Reviewer_2: The authors also studied the effect of CT and Ft in lower limb strength, however, there isn’t any justification for it in the introductory section.
Response: Dear reviewer, the aim of also assessing adaptations in muscle strength stems from our concern to check whether morphological adaptations are accompanied by functional adaptations. We have chosen the five times sit-to-stand test because it is indicated in the latest International Consensus on Sarcopenia as a method for assessing muscle strength (10.1093/ageing/afy169). We also believe that this needs to be clear to the Healthcare reader, so we have made some adjustments to the introduction. (lines 85 – 88)
#Reviewer_2: Methods
The authors identified in the intervention calendar a familiarisation phase. It will be interesting to identify what the authors did in this phase.
Response: Dear reviewer, we have adjusted the initial sentence in which we quoted familiarization (line 109 – 110) and inserted more specific information in the "intervention protocols" section (lines 151 – 154)
#Reviewer_2: Line 115 - These are the conditions for women to be in the menopausal phase. These criteria are indicated for whom?
Response: Dear reviewer, thank you for your comment. Our inclusion criteria include the age of the woman, as well as the lack of periods in the last 12 months. This categorization of "postmenopausal" is in line with the proposal published in the journal "Menopause: The Journal of The North American Menopause Society" (DOI: 10.1097/01.GME.0000141980.81159.B2). It is also in line with the proposals used by the World Health Organization (https://www.who.int/news-room/fact-sheets/detail/menopause), National Library of Medicine (https://www.ncbi.nlm.nih.gov/books/NBK507826/) and National Institute on Aging (https://www.nia.nih.gov/health/menopause/what-menopause).
#Reviewer_2: Line 137 – “The participants performed all exercises at maximum concentric speed. To control the external load, we maintained a range between 8 and 12 repetition maximum (RM)…” 8 and 12 repetitions maximum? Please, clarify.
Response: Dear, it’s done. Thank you. (Lines 155 – 161)
#Reviewer_2: Line 144 – “The intervention protocols were performed at the gym of the institution.” What institution?
Response: The information has been duly inserted (lines 166 – 167). Thank you.
#Reviewer_2: Line 162 – Why the control group performed submaximal static stretching exercises with two sets of 15 seconds for the major muscle groups and meditation practices?
Response: Dear reviewer, this training proposal is intended to provide the participants with a non-pharmacological health promotion strategy. In our opinion, it would be unethical to maintain a group of older women during 16 weeks without any exercise, given the negative health effects of sedentary behavior. In addition, it is valid to say that the control group, after the end of the study, also received the best available intervention. In this case, they chose CT or FT.
From a scientific point of view, it is important to point out that this intervention (submaximal static stretching exercises) cannot promote adaptations in body composition and muscle strength, as was found in the results. In this way, do not influence the findings and outcomes of the study.
#Reviewer_2: Results
Line 227 - It is the first time that the term appendicular lean mass appears. Please, uniform all document.
Response: Dear reviewer, we have reinforced in the methods section that this variable was assessed by the bioimpedance system, being a specific variable of body composition (lines 199 – 200). Thank you.
#Reviewer_2: Discussion
It will be very interesting to have some discussion about the identical variations between FT and CT.
Response: Dear reviewer, we have inserted changes in the text to make this discussion clearer (lines 291 – 293). Thank you.
#Reviewer_2: Conclusion
Line 317 – Please, fix - similar cts in.
Response: Done, thank you. (Line 344)
We would like to thank you for your contribution and we are grateful for the new version of the article that has been achieved as a result of the changes provided by your comments. We hope that the answers sent, as well as the changes derived from the review process, address your concerns and meet the necessary requirements for publication.
Reviewer 3 Report
Comments and Suggestions for Authors
We thank you for the opportunity to review your manuscript.
Please respond to the following comments for publication.
1. The significance of this study should be indicated at the end of the introduction.
2. Describe how you recruited participants for this study.
3. You mentioned that you monitor them during the intervention, but how do you ensure reliability with respect to their diet and daytime activities? For example, do they keep a record of their meals or use an activity meter? We believe that this issue needs attention, especially in older adults. This is because the reliability of records is uncertain due to cognitive decline and other factors.
4. The reliability and validity of the machine that measured body composition should also be described.
5. Is it possible that past exercise habits and other factors may influence the results, but are they not being evaluated?
Author Response
#Reviewer_3: We thank you for the opportunity to review your manuscript.
Please respond to the following comments for publication.
Response: Dear reviewer, thank you for your notes. The changes emerged by our conversation were highlighted in green in the main document.
#Reviewer_3: 1. The significance of this study should be indicated at the end of the introduction.
Response: Dear reviewer, we have made some changes to the text to make this significance clearer (lines 94 – 97). Thank you.
#Reviewer_3: 2. Describe how you recruited participants for this study.
Response: Dear reviewer, thank you for your comment. We have detailed this recruitment in the text (lines 122 – 124).
#Reviewer_3: 3. You mentioned that you monitor them during the intervention, but how do you ensure reliability with respect to their diet and daytime activities? For example, do they keep a record of their meals or use an activity meter? We believe that this issue needs attention, especially in older adults. This is because the reliability of records is uncertain due to cognitive decline and other factors.
Response: Dear reviewer, thank you for your comment. We understand the importance of physical activity level and diet in our study, but a limitation of our study was that we did not conduct any form of monitoring of these factors. We understand that the most classic forms of nutritional and physical activity assessment, carried out by interview, can be biased when applied to this target audience, as rightly pointed out by the reviewer.
However, at the beginning of the intervention, we asked all participants to avoid changing their habitual physical activity and diet routines during the intervention period. In addition, we reinforce that other studies have found adaptations to body composition without control of diet and physical activity (DOI: 10.1016/j.pcad.2018.07.014), just as we also found. This can be explained by the tendency of older adults to maintain healthy eating behaviors (PMID: 11426286) (10.1016/j.maturitas.2013.05.005).
Finally, we would like to emphasize that this approach is more closely aligned with conditions resembling real-world scenarios when there is no access to the nutritional and physical activity aspects of the participants.
We have properly discussed this limitation for the Healthcare readers (lines 325 - 332).
#Reviewer_3: 4. The reliability and validity of the machine that measured body composition should also be described.
Response: Dear reviewer, thank you for your comment. We carried out a prior assessment (n = 30) to ensure good reliability indicators. We find an intraclass correlation coefficient (ICC) bidirectional mixed effects type, absolute agreement, single measurement with a value of 0.91, considered excellent in the literature. Furthermore, a previous study (DOI: 10.1590/S1806-37562017000000121) showed very high levels of agreement between the bioimpedance system used and the gold standard model in the literature (Dual-energy X-ray absorptiometry). This information was included in the manuscript file (lines 208 – 212).
#Reviewer_3: 5. Is it possible that past exercise habits and other factors may influence the results, but are they not being evaluated?
Response: Dear reviewer, it is plausible that long-life exercise habits before the intervention program could impact the results. However, we tried to minimize this possible influence as much as possible with our inclusion criterion "have not exercised regularly for at least six months". In this sense, the baseline values do not present a difference between the groups, denoting the success of the randomization process. In that regard, we believe that six months is enough time for there to be a washout in variables related to body composition, as demonstrated in previous studies (DOI: 10.2147/CIA.S299867 and DOI: 10.1016/j.jbmt.2023.11.026).
We would like to thank you for your contribution and we are grateful for the new version of the article that has been achieved as a result of the changes provided by your comments. We hope that the answers sent, as well as the changes derived from the review process, address your concerns and meet the necessary requirements for publication.
Round 2
Reviewer 2 Report
Comments and Suggestions for Authors
Dear authors,
Thank you for addressing my comments.
Best regards,
Reviewer 3 Report
Comments and Suggestions for Authors
Author has made careful revisions in response to the comments.